# Monthly Radial Growth Model of Chinese Fir (*Cunninghamia lanceolata* (Lamb.) Hook.), and the Relationships between Radial Increment and Climate Factors

**Yaqi Huang [1], Xiangwen Deng [1,2,3,*], Zhonghui Zhao [1,2,3], Wenhua Xiang [1,2,3], Wende Yan [1,2,3], Shuai Ouyang [1,2,3] and Pifeng Lei [1,2,3]**

1    Faculty of Life Science and Technology, Central South University of Forestry and Technology, Changsha 410004, China
2    Huitong National Field Station for Scientific Observation and Research of Chinese Fir Plantation Ecosystem in Hunan Province, Huitong 438107, China
3    National Engineering Laboratory for Applied Technology of Forestry & Ecology in South China, Changsha 410004, China
*    Correspondence: dengxw@csuft.edu.cn; Tel.: +86-731-85623458; Fax: +86-731-85623458

**Abstract:** Chinese fir (*Cunninghamia lanceolata* (Lamb.) Hook) is the most commonly grown afforestation species in subtropical China. It is essential that we understand the response of radial tree growth to climate factors, yet most experiments have been conducted based on total annual growth and not on monthly dynamics, which alone can detail the influence of climatic factors. In this study, we aimed to: (i) construct a monthly growth model and compare the growth rate of different social statuses of trees, and (ii) determine the response of radial increments of different social statuses to climate factors. The radial growth was monitored monthly during four years using manual band dendrometers (MBD). The data were fitted using the Gompertz function. Within-stand differences in the social status of Chinese firs resulted in growing period and growth rate length variations. The radial growth began in March, and suppressed trees—especially groups of AS1 and BS1 (suppressed trees of classes I in sites A and B)—stopped in September, whereas dominant and intermediate trees were delayed and stopped in November. The periodic monthly increment curve showed double peaks, and the maximum growth rate occurred in April and August. The peak values were affected by social status, which showed that dominant trees had the greatest radial growth rates. S-shaped Gompertz meant that monthly increment models were successfully fitted to our data, which explained more than 98% of the variation in increment data and passed the uncertainty test. Temperature and precipitation had a significant influence on radial growth, and the correlation between radial growth and air temperature was the highest. Our results also revealed that temperatures explain the double-peak features of Chinese fir. The limiting factors of radial growth changed with the seasons and were mainly affected by temperature and precipitation, which should be considered in predicting the response of tree growth to climate change.

**Keywords:** climate change; double peaks; Gompertz model; manual band dendrometer; radial growth; subtropical area

## 1. Introduction

There is an expectation that many species will be affected by climate change in the 21st century [1]. As the largest component of the terrestrial biosphere, the forest ecosystem plays a key role in the

soil–plant–atmosphere continuum [2]. Tree growth is not only related to the trees' own biological characteristics, such as species and social status [3–6], but also affected by topographic and climate factors [7–11]. Thus, it is necessary to investigate tree growth and its response to climate change for protecting forests and the ecosystem, which provides benefits to society [12,13].

Alterations in the width of tree rings are known to encode important climate factors, such as air temperature and precipitation. To discover the details, seasonal radial growth patterns are needed in different forest ecosystems [14]. Information on the high resolution (short timescale) tree-stem increments may be lost using the dendrochronology method, it is thus necessary to introduce monitoring methods to explore the dynamic variation of monthly radial growth. The dynamic variation in tree diameter can change over periods of time ranging from hours to years, which provides an important basis for characterizing multiple aspects of tree performance and forest–microclimate interactions [6]. Based on seasonal variation, trees can reveal some important events in the process of radial growth, such as when tree growth began or stopped [15].

In temperate and boreal forests, cumulative radial growth was observed as an S-shaped curve, and there were some variations among tree species [16–18]. As a result, non-linear and parametric curve-fitting provisions were widely used to understand diverse ecological and evolutionary trends [19]. When using a non-linear growth curve model, the inflection point indicated the maximum rate of tree growth [20]. On the curve of the Gompertz model, the position of the inflection point appears early. Hence, it is suitable for fitting growth trends of fast-growing species. In a previous study, the S-shaped Gompertz model was used to estimate daily growth rates and relative periods of differentiation growth phases [16]. The Gompertz model has been applied to a large range of studies relating to the seasonal radial stem dynamic of different species of trees, such as continental and oceanic temperate forests [5,18], Mediterranean forests [21,22], alpine forests [23,24], and boreal tree species [16,25,26].

The subtropical area of South China is considered to be an important region for biodiversity and a great nature reserve for endemic plant species. However, it is considered an important vulnerable zone of climate change [27]. In subtropical regions, tree radial growth studies are considered main hot-spot research for understanding the climate–growth relationship [28]. Chinese fir (*Cunninghamia lanceolata* (Lamb.) Hook.) is a fast-growing dominant conifer tree species in subtropical China [29,30]. Due to the high quality of timber, this species plays an important role in terms of timber supply—multiple forest by-products for human society. In addition, Chinese fir promotes environmental protection and maintains balance in the forest ecosystem [31–33]. This fast-growing tree is able to survive in an environment with abundant annual precipitation and warm temperature [34]. For growth study in a subtropical region, radial growth of tree research is fundamental for understanding the climate–growth relationship, which can directly show the response of tree growth to climate change [28]. However, studies on the relationship between growth and climate factors have been mostly focused on inter-annual changes. The dynamics of monthly radial growth are still poorly understood.

In this study, the monthly radial increment data of Chinese fir were monitored for four years using a manual band dendrometer (MBD), and were modelled using the Gompertz function, to comprehensively understand the impacts of climate on tree growth by combining traditional climatic data, such as temperature, precipitation, and relative humidity. Our objectives were: (i) to construct a monthly growth model and compare the growth rate of different social statuses of trees, and (ii) to determine the response of radial increment to climate factors. The main hypotheses were: (i) the period and rate of radial growth change according to social status within the stand, and (ii) the sensitivity of radial growth to air temperature is higher than that to precipitation and relative humidity.

## 2. Materials and Methods

### 2.1. Study Area

The study site was the Huitong National Field Station for Scientific Observation and Research of Chinese Fir Plantation Ecosystem (109°35′—109°36′ E, 26°46′—26°48′ N) in Hunan province, China. In

this region, the elevation ranges from 270 m to 400 m above sea level, and the area experiences a typical subtropical monsoon climate which is characterized by mild winters and hot summers, and the rainy season occurs from May to October. The whole year's growth period is more than 300 days, which is conducive to the growth of Chinese fir. Long-term records (1951–2018) from the meteorological station in Huitong (26°47' N, 109°38' E, 308 m a.s.l.) showed that the average annual precipitation was 1337 mm. May and June are the wettest months, and December is the driest month. The annual mean temperature is 16.6 °C (Figure 1). The soil type is mountain loessial soil, its texture is between light soil and medium clay soil, and the main layer's thickness is over 80 cm. The pH is ~4.86. The soil is identical on both sites. The plant vegetation surrounding the station is a typical subtropical evergreen broad-leaved species. There are only a few shrubs and herbs under the forest. The representative undergrowth vegetation is mainly composed of *Maesa japonica* (Thunb.) Moritzi., *Woodwardia japonica* (L. f.) Sm., and *Dicranopteris linearis* (Burm.) Underw. [35].

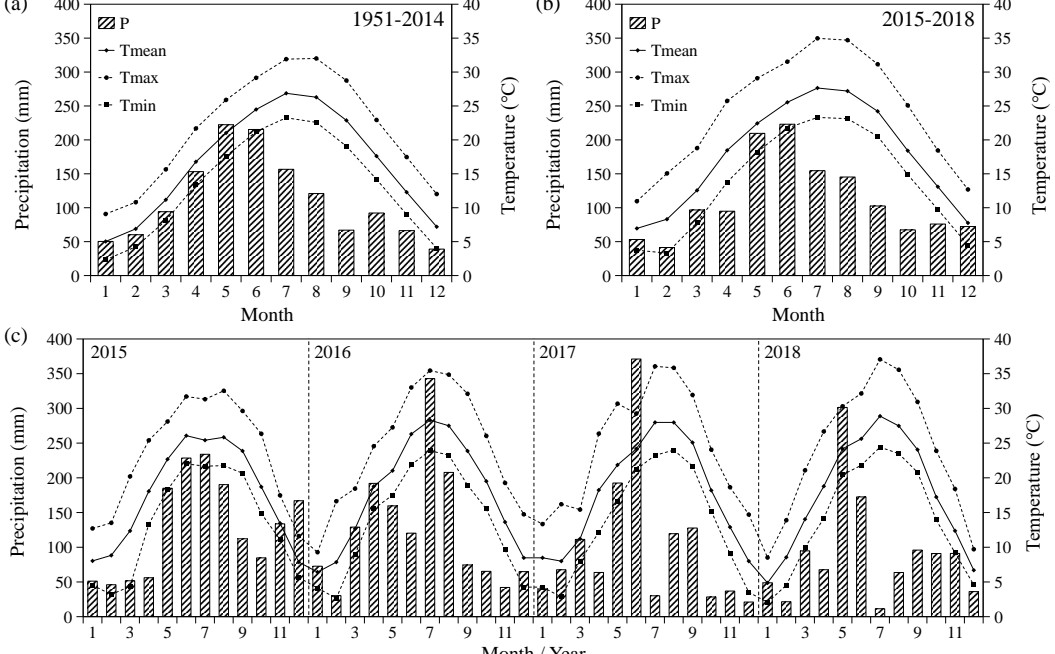

**Figure 1.** Monthly precipitation (P), monthly mean (Tmean), average of maximum (Tmax), and minimum (Tmin) temperatures (°C) of Huitong from 1951 to 2014 (**a**), from 2015 to 2018 (**b**), and from January 2015 to December 2018 (**c**), recorded at our ecological research station in Huitong, Hunan province.

### 2.2. Sample Plot Selection

We selected two sample plots (20 m × 30 m) from the station as our study sites (site A and B). Two plots were clear cut, burned, and artificially planted in 1988, and second generation of Chinese fir plantations were established. In spring 1996, site A was reforested for the low survival rate, and site B remained undisturbed. As a result, the ages of sample trees were different in the two sites, trees in sites A and B were 22 and 30 years in 2018, respectively. As our experimental forest, there was no human disturbance.

### 2.3. Tree Increment Data

In January 2015, we measured the diameter at breast height (DBH), tree height, crown width, and spatial position of each tree in the sample plot. At the same time, we also investigated the basic information of the sample plots. In addition, we selected 162 well-grown Chinese firs and installed MBDs at breast height (Figure 2). The specific installation methods and working principles followed



Vitas [36]. We used a vernier caliper to measure the slot length of each MBD within 0.01 mm accuracy on the first day of every month.

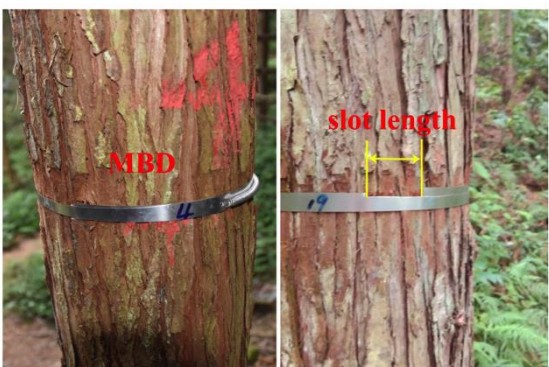

**Figure 2.** Manual band dendrometer (MBD) installed at breast diameter of Chinese fir.

*2.4. Climate Data*

The climate factors of air monthly mean, maximum and minimum temperatures (°C), relative humidity (RH, %), and monthly precipitation (P, mm) were considerable or were used in this study. The 20 m height meteorological observation tower was set up in the sample plot. There is probe for measuring hourly air temperature and relative humidity (HMP45C-L, Campbell Scientific, Inc., Logan, UT, USA) at a height of 2 m above the crown. The siphon rainfall recorder was used for continuous recording of atmospheric rainfall. To harmonize the time resolution between MBD and climate data, the daily climate data were processed into monthly averages or sums of climate data.

*2.5. Data Analysis*

2.5.1. Sample Grouping

Each sample tree was grouped according to the site and basal area of breast height. According to the order of diameter classes and the average growth rate in these four years, the sample trees were divided into five diameter classes with the same basal area. Classes I and II were suppressed trees, which included groups AS1, AS2, BS1, and BS2; class III was intermediate trees including AI1 and BI1; classes IV and V were dominant trees, including AD1, AD2, BD1, and BD2. Information regarding the 10 groups' sampling trees in two sites are shown in Table 1.

**Table 1.** Information of observed sample trees and monitoring sites.

| Site | Groups | Number | Age (years) | Slope (°) | Slope Aspect | Average DBH | Max DBH | Min DBH | Variance |
|------|--------|--------|-------------|-----------|--------------|-------------|---------|---------|----------|
| A | AS1 | 22 | 22 | 25 | Northeast | 15.32 | 18.93 | 12.50 | 4.96 |
| A | AS2 | 20 | 22 | 25 | Northeast | 18.10 | 23.23 | 14.26 | 4.72 |
| A | AI1 | 17 | 22 | 25 | Northeast | 21.73 | 23.98 | 19.21 | 1.61 |
| A | AD1 | 14 | 22 | 25 | Northeast | 23.56 | 25.87 | 20.05 | 1.84 |
| A | AD2 | 9 | 22 | 25 | Northeast | 25.63 | 26.72 | 24.26 | 0.74 |
| B | BS1 | 23 | 30 | 27 | Northeast | 17.12 | 19.85 | 13.55 | 2.37 |
| B | BS2 | 20 | 30 | 27 | Northeast | 20.63 | 22.64 | 17.22 | 2.72 |
| B | BI1 | 16 | 30 | 27 | Northeast | 24.11 | 26.99 | 19.72 | 2.73 |
| B | BD1 | 12 | 30 | 27 | Northeast | 26.85 | 33.40 | 25.40 | 4.63 |
| B | BD2 | 9 | 30 | 27 | Northeast | 29.39 | 36.49 | 26.50 | 8.69 |

DBH: diameter at breast height; AS1: suppressed trees of classes I in site A; AS2: suppressed trees of classes II in site A; AI1: intermediate trees of classes III in site A; AD1: dominant trees of classes IV in site A; AD2: dominant trees of classes V in site A; BS1: suppressed trees of classes I in site B; BS2: suppressed trees of classes II in site B; BI1: intermediate trees of classes III in site B; BD1: dominant trees of classes IV in site B; BD2: dominant trees of classes V in site B.

### 2.5.2. Growth Pattern and Rate

The slot increment was translated into units of DBH for further analyses. The slot length measured with vernier calipers was linear distance instead of arc distance, so we wrote a program code to eliminate error and to calculate the monthly DBH increment. The monthly growth rate was calculated using the following equation:

$$\mu_t = \frac{g(t) - g(t_{-1})}{t - t_{-1}} \tag{1}$$

where $t$ is the day of the year (DOY), $t_{-1}$ is the sampling date before $t$, and $g(t)$ and $g(t_{-1})$ are the radial increments at times $t$ and $t_{-1}$, respectively.

### 2.5.3. Growth Curves Modelling

The radial growth was fitted using a non-linear Gompertz model estimation from the 'grofit' packet in R 3.5.2 software [37] according to the following formula:

$$y(t) = Ae^{-e[\frac{\mu e}{A}(\lambda - t) + 1]} \tag{2}$$

where y is the monthly cumulative radial increment (µm), $t$ is the day of year (DOY), $A$ is the upper asymptote (µm), fixed according to the maximum growth, $\mu$ is maximum growth rate, and $\lambda$ is lag phase. Based on a model-based fit and a model-free spline fit, we used two different methods to fit a given growth curve [38]. In order to extract parameters from different social statuses, each group was fitted with Gompertz model and smoothing spline. The extracted growth parameters ($A$, $\mu$, and $\lambda$) from trees were analysed by two-way analysis of variance (ANOVA) to compare the differences between sites and groups and inter-annual differences between 2015 and 2018. Uncertainty analysis of model fitting was also considered.

### 2.5.4. Relationship between Climate and Radial Growth

The stepwise regression and Pearson correlation coefficients were calculated to quantify the relationship between climate factors and the monthly radial growth. Analysis of covariance (ANCOVA) was used to compare the effects of climate factors on radial increment over four years and among different social statuses. Moreover, the regression between climatic factors and monthly radial increment was established. All statistical analyses were conducted using R software (3.5.2), and all differences were significant at $p < 0.05$.

## 3. Results

### 3.1. Monthly Radial Growth

In both sites, the monthly pattern of stem radial variations showed double-peak curves during the tree-growing period, although there were differences in growth patterns among different diameter classes (Figure 3). With a change of social status, the bigger the diameter class, the more obvious the bimodal pattern, especially in dominant trees. The radial growth of dominant and intermediate trees began in March and stopped in November, whereas suppressed trees, especially groups AS1 and BS1, stopped in September. The radial growth rate progressively increased in April, followed by a marked decrease or plateau in June and July, and then a sharp increase in August. It started to stabilize in November. The growth in April was obviously higher than that in August. However, there were some differences in the maximum growth rates in different diameter classes, and the peak values of radial growth rates of dominant trees were higher than that of suppressed trees.

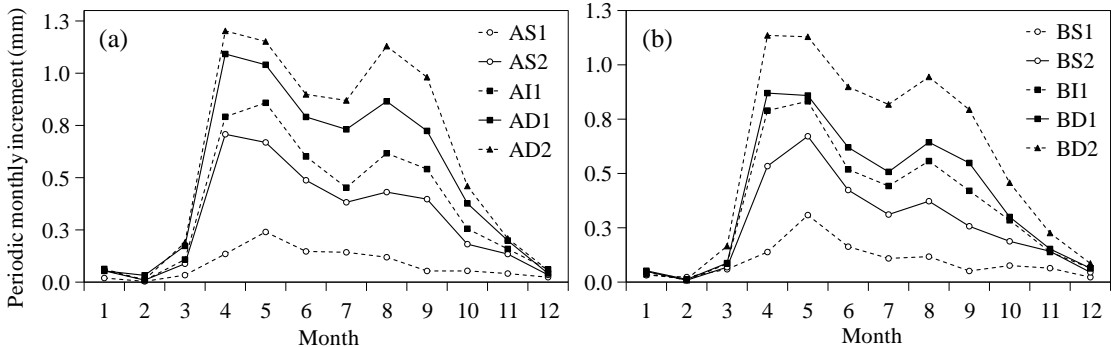

**Figure 3.** Average month radial growth increment of the five groups of trees at site A (**a**) and site B (**b**).

### 3.2. Growth Curves' Modelling

Gompertz models explained more than 98% of the variation in monthly radial increment data (Figure 4), indicating that the model was suitable for radial growth of Chinese fir. The means' monthly increment curves exhibited an S shape across a whole year. The mean monthly increment of dominant trees in site A was larger than that in site B, whereas the growth of suppressed trees in site A was smaller than that in site B. The difference of maximum increment among the social status of 22-year-old trees was larger than that of 30-year-old trees. During 2015–2018, the trees' annual radial increment in 2018 was smaller than in other years.

The fitting parameters were exported, $\lambda$ changed greatly, while $A$ and $\mu$ changed slightly, the maximum values of $A$ and $\mu$ both appeared in 2016 in a four-year comparison, and the minimum values of parameters ($A$, $\mu$ and $\lambda$) appeared in 2018 (Table 2). Across the two sites, $A$ and $\mu$ increased with the increasing tree diameter class. In addition, two-way ANOVA analysis showed that $A$ and $\mu$ of different diameter classes had significant difference in both site, and $A$ and $\lambda$ were significantly different in inter-annual variation ($p < 0.001$) (Table 3).

**Table 2.** Mean and standard deviation (SD) of radial growth and shape parameters for the radial growth of Chinese fir.

| Items | | $A$ | | $\mu$ | | $\lambda$ | |
|---|---|---|---|---|---|---|---|
| | | Mean | SD | Mean | SD | Mean | SD |
| Year | 2015 | 4.901 | 0.140 | 0.025 | 0.001 | 97.627 | 4.180 |
| | 2016 | 4.959 | 0.146 | 0.026 | 0.002 | 87.183 | 4.957 |
| | 2017 | 4.470 | 0.168 | 0.025 | 0.002 | 82.967 | 6.833 |
| | 2018 | 3.592 | 0.161 | 0.021 | 0.002 | 81.016 | 8.784 |
| Group | AS1 | 1.027 | 0.015 | 0.006 | 0.000 | 88.612 | 2.964 |
| | AS2 | 3.743 | 0.095 | 0.021 | 0.001 | 82.675 | 4.627 |
| | AI1 | 4.791 | 0.134 | 0.025 | 0.001 | 85.232 | 4.763 |
| | AD1 | 6.572 | 0.174 | 0.034 | 0.002 | 86.121 | 4.400 |
| | AD2 | 7.893 | 0.255 | 0.040 | 0.002 | 90.275 | 4.953 |
| | BS1 | 1.175 | 0.030 | 0.007 | 0.000 | 76.824 | 5.201 |
| | BS2 | 3.161 | 0.076 | 0.018 | 0.001 | 84.360 | 4.521 |
| | BI1 | 4.391 | 0.125 | 0.024 | 0.001 | 84.708 | 5.021 |
| | BS1 | 5.009 | 0.143 | 0.027 | 0.002 | 86.255 | 4.818 |
| | BS2 | 7.194 | 0.186 | 0.038 | 0.002 | 89.684 | 4.162 |

$A$: maximum growth; $\mu$: maximum growth rate; and $\lambda$: lag time of growth.

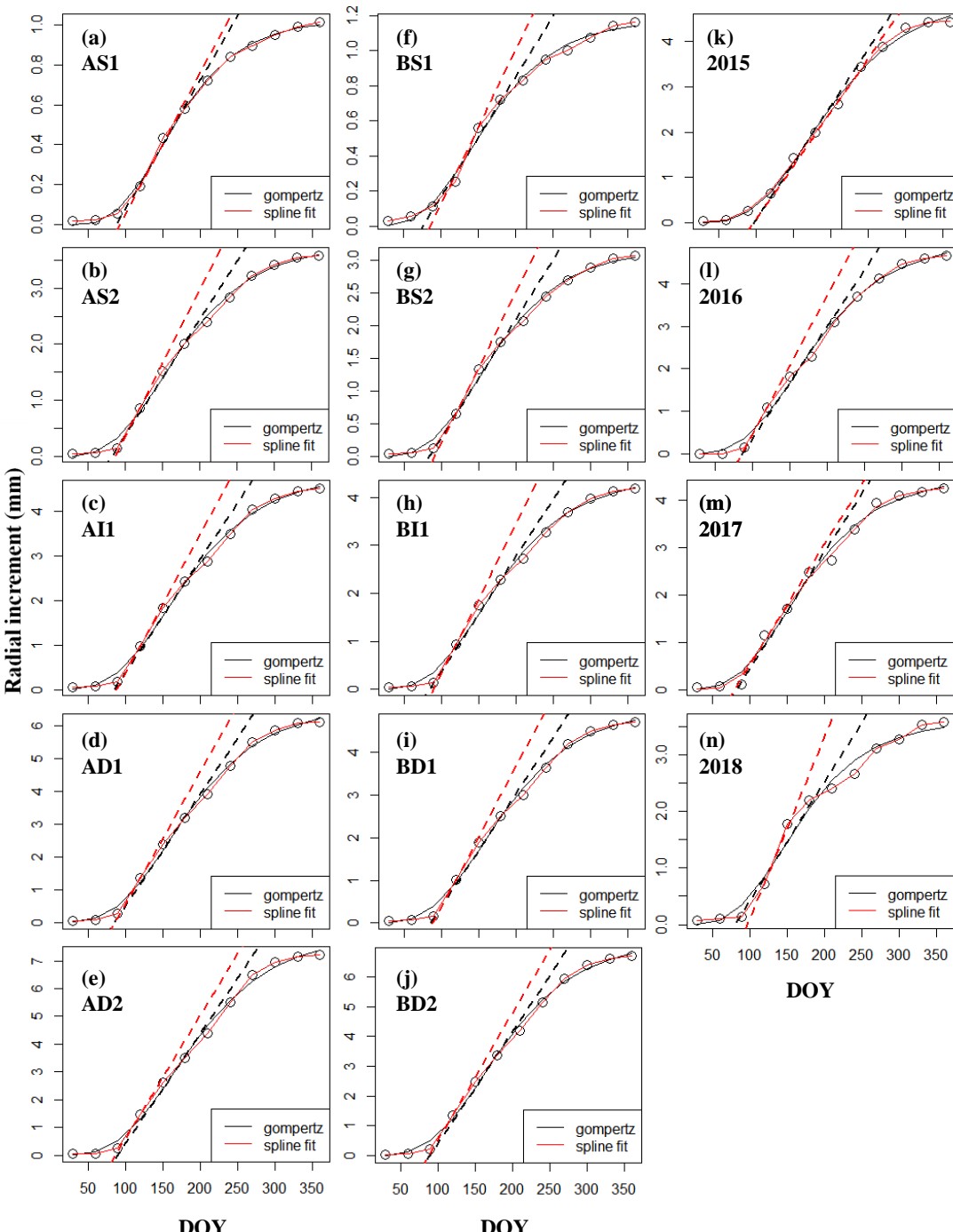

**Figure 4.** Estimated radial growth curves by the Gompertz model for Chinese fir (solid line) and the maximum slope (dash line) throughout a year: (**a**–**e**) Considering the tree groups in site A, (**f**–**j**) considering the tree groups in site B, and (**k**–**n**) considering the studied years. Note: DOY means day of the year.

**Table 3.** Two-way analysis of variance (ANOVA) of the effects of tree group and the inter-annual differences of the growth and parameters $A$, $\mu$, and $\lambda$.

| Factors Site A | A | | $\mu$ | | $\lambda$ | |
|---|---|---|---|---|---|---|
| | F | p | F | p | F | p |
| Year | 9.472 | 0.002 ** | 1.339 | 0.308 | 19.989 | <0.001 *** |
| Group | 416.193 | <0.001 *** | 201.847 | <0.001 *** | 0.009 | 0.926 |
| Year × Group | 2.908 | 0.078 | 1.030 | 0.414 | 0.774 | 0.531 |
| **Factors Site B** | A | | $\mu$ | | $\lambda$ | |
| | F | p | F | p | F | p |
| Year | 7.298 | 0.005 ** | 4.241 | 0.029 * | 6.371 | 0.008 ** |
| Group | 351.113 | <0.001 *** | 234.306 | <0.001 *** | 9.730 | 0.009 ** |
| Year × Group | 4.452 | 0.025 * | 2.316 | 0.128 | 2.402 | 0.119 |

$A$: maximum growth; $\mu$: maximum growth rate; and $\lambda$: lag time of growth. * $p < 0.05$; ** $p < 0.01$; *** $p < 0.001$.

### 3.3. Radial Increment Responses of Climate Factors

Stepwise regression analysis demonstrated that temperature had a significant effect on stem growth throughout the whole growing season (Table 4). In the optimal regression equation, all regression equations were significant ($p < 0.001$). Climate factor of the monthly average for minimum temperature was excluded in sites A and B. For the residual test, the residuals vs fitted graph was random distribution, and the points in the normal Q–Q graph basically fell on a straight line, indicating that the residual obeys normal distribution. Therefore, the significance test of stepwise regression showed that the selected independent variables had a significant effect on stem increment, and the constructed regression equation was optimal.

**Table 4.** Stepwise regression of radial growth and climate factors in different groups.

| | Intercept | Tmean | P | Tmax | Tmin | RH |
|---|---|---|---|---|---|---|
| AS1 | −1.0648 * | −0.0263 | 0.0003 ** | 0.0303 | - | 0.0101 * |
| AS2 | −3.0882 * | −0.1251 * | 0.0010 * | 0.1416 * | - | 0.0274 |
| AI1 | −3.0692 | −0.1256 | 0.0012 * | 0.1480 * | - | 0.0265 |
| AD1 | −3.5633 | −0.1499 | 0.0015 ** | 0.1832 * | - | 0.0296 |
| AD2 | −0.4898 *** | - | 0.0020 *** | 0.0430 *** | - | - |
| BS1 | −1.1378 | −0.0478 * | 0.0003 * | 0.0506 * | - | 0.0107 |
| BS2 | −3.0093 * | −0.1305 * | 0.0008 * | 0.1422 ** | - | 0.0274 * |
| BI1 | −3.5720 * | −0.1618 * | 0.0010 * | 0.1796 ** | - | 0.0318 |
| BD1 | −3.6035 | −0.1371 | 0.0011 * | 0.1601 * | - | 0.0324 |
| BD2 | −3.5852 | −0.1415 | 0.0017 ** | 0.1762 * | - | 0.0304 |

Tmean: monthly mean temperature; P: monthly precipitation; Tmax: monthly average of maximum temperature; Tmin: monthly average of minimum temperature; and RH: relative humidity. * $p < 0.05$; ** $p < 0.01$; *** $p < 0.001$.

The correlation analysis showed that radial growth was significantly correlated with air temperature and precipitation ($p < 0.001$) (Figure 5). The correlation between radial growth of tree and air temperatures was the highest, and dominant trees had more sensitivity to air temperature ($R_{a5}^2 = 0.74$, $p < 0.001$; $R_{b5}^2 = 0.76$, $p < 0.001$). Monthly average of maximum temperature greatly affected radial growth, and the correlation increased with the increasing of diameter classes. Furthermore, stem growth was also significantly affected by precipitation ($p < 0.001$), and weakly responded to relative humidity. There was no significant difference in the response of stem variation to climate factors between the two sites. No obvious differences between monthly average of maximum, minimum, and mean temperatures on the radial growth of Chinese fir were observed, therefore, the effect of mean temperature was only discussed in this paper. In spring, autumn, and winter, the correlation

coefficient between temperature and radial growth is positive, but in summer, the relationship is negative (Figure 6). The ANCOVA result showed significant interaction between groups and climate factors ($p < 0.001$), suggesting that temperature and precipitation differentially regulate the radial growth of dominant and suppressed trees (Table 5).

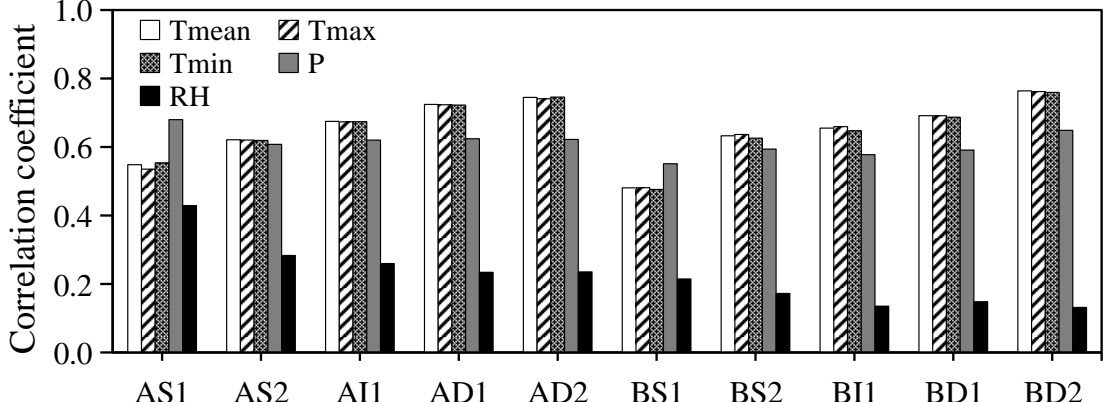

**Figure 5.** Correlations between radial growth and climate factors in two sites. Tmean is monthly mean temperatures, Tmax is monthly average of maximum temperature, Tmin is monthly average of minimum temperature, and P is precipitation.

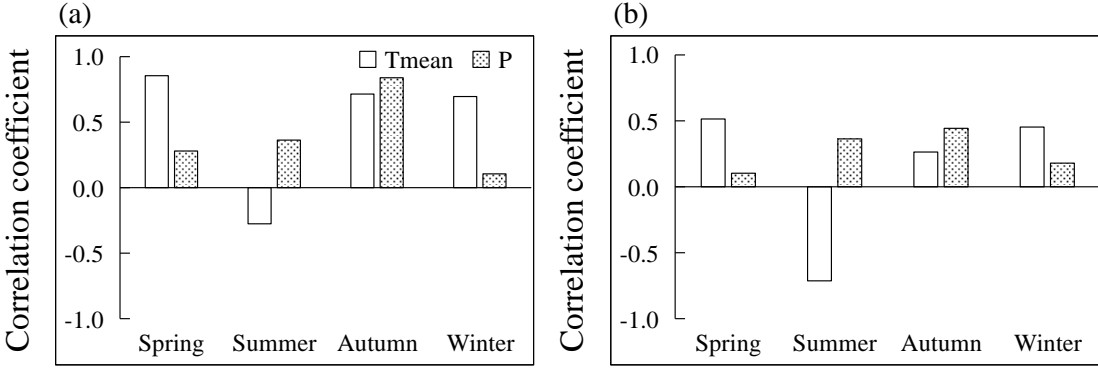

**Figure 6.** Correlation coefficients between seasonal temperatures (Tmean), precipitations (P), and radial growth of group AD2 at site A (**a**) and BS1 at site B (**b**).

**Table 5.** Analysis of covariance (ANCOVA) of climatic factors on radial growth.

| Items | Site A | | Site B | |
|---|---|---|---|---|
| | *F* | *p* | *F* | *p* |
| Temperature | 186.92 | <0.001 *** | 182.44 | <0.001 *** |
| Group | 27.38 | <0.001 *** | 26.22 | <0.001 *** |
| Temperature × Group | 12.41 | <0.001 *** | 12.24 | <0.001 *** |
| Precipitation | 120.72 | <0.001 *** | 113.99 | <0.001 *** |
| Group | 21.99 | <0.001 *** | 20.98 | <0.001 *** |
| Precipitation × Group | 5.99 | <0.001 *** | 6.27 | <0.001 *** |

*** $p < 0.001$.

With an increase in temperature (Figure 7a,b) and precipitation (Figure 7c,d), monthly growth increased rapidly. The regression line's slope between radial growth and relative humidity was smaller (Figure 7e,f). Compared to the precipitation ($R^2 = 0.261$ in site A and $R^2 = 0.252$ in site B) and relative humidity ($R^2 = 0.045$ in site A and $R^2 = 0.015$ in site B), the radial growth of Chinese fir was more

sensitive to temperature ($R^2 = 0.324$ in site A and $R^2 = 0.322$ in site B). It was found that the correlation coefficient between sample tree radial increment and climate factors were higher at site A than at site B.

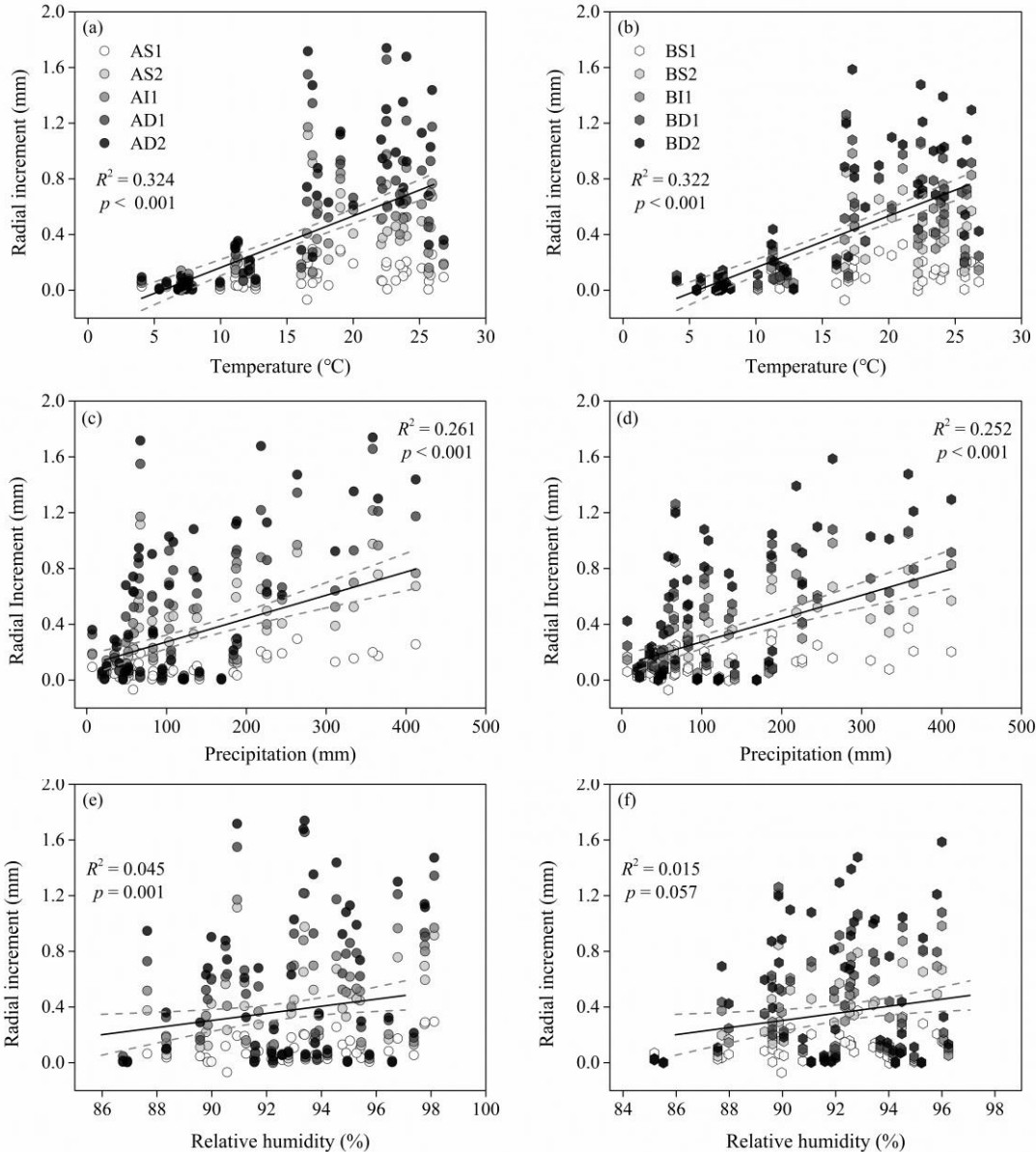

**Figure 7.** Linear relation of climate factors and radial growth. (**a**) Temperature and growth in site A, (**b**) temperature and growth in site B, (**c**) precipitation and growth in site A, (**d**) precipitation and growth in site B, (**e**) Relative humidity (RH) and growth in site A, and (**f**) RH and growth in site B. The grey dashed line represents 95% confidence interval (CI).

## 4. Discussion

### 4.1. Influence of Trees' Social Status on Radial Growth

Our research shows differences in the growth period and monthly increment of different social statuses (Figure 3). Confirming our first hypothesis, evidence indicated that the growing season of dominant trees was longer, whereas that of suppressed trees was shorter, especially groups of AS1 and BS1, and periodic monthly increments of dominant trees were higher than other trees. There is a deterministic relationship between niche overlap and inter-species competition, which may influence tree growth of different social statuses [39]. Tree species with a large DBH usually have bigger canopies

and are able to obtain more water, nutrients, and sunlight when competing with small-sized trees, leading to higher growth rate and periodic monthly increments. The intra-annual dynamic growth of trees was affected by social status (dominant or suppressed), leading to varying responses of trees to environmental conditions [5].

Previous research indicated a strong relationship between the beginning, end, and duration of cambial activity and the status of a tree, the periodicity of cambial activity lasting longer for dominant trees [5,21,40]. This research on conifer species was consistent with the present findings that the growing period of suppressed trees was shorter than that of dominant trees. Such a difference in growth periodicity should lead to many other important differences in life form, function, and the adaptability of trees. Therefore, the climate–growth relationship for dominant trees and suppressed trees was different [5]. For the specific activities of cambium and detailed information of xylem formation, especially accurate timing and duration of cambium activity, in the growing period of Chinese fir, further studies at the cellular level are needed.

### 4.2. Effects of Climate Factors on the Double-Peaks Pattern

Temperature is a primary factor affecting the radial growth of trees in boreal, sub-alpine, and temperate forests [18,41,42]. In subtropical areas, we found that the effect of air temperature on stem growth was generally higher than that of precipitation (Figures 5 and 7), confirming the assumption that the radial increment of Chinese fir is more sensitive to temperature. It has been demonstrated that when the critical temperature is reached, the production of coniferous xylem cells tends to reactivate in cold temperatures [43]. There are two reasons for rapid radial growth in plants with increased temperature [44]. Firstly, with the increasing temperature, the ability of tree leaves to photosynthesize and carbon fixate is increased [45]. Secondly, increasing the temperature also enhances the activity of enzymes related to photosynthesis in leaves, thus improving the photosynthetic ability of leaves and further promoting stem radial growth [46].

The dynamics of stem radial growth is limited by the double climatic stress of cold winters and hot summers in subtropical regions. In subtropical regions, summer climate is the key factor for tree radial increment [28]. The climate of China's subtropical region is characterized by East Asian monsoon, which is mild in winter and hot in summer. Suppression of summer growth seems to be a strategy for coping with harsh environmental conditions during summer drought [47]. The growth rate of Chinese fir decreased in the summer (June and July), showing a bimodal growth pattern (Figure 3). Temperatures in the spring promoted the radial growth of plants, whereas high temperatures in the summer inhibited the growth rate (Figure 6). High temperatures in the summer significantly affected the transpiration of vegetation and the osmotic adjustment of energy metabolism and respiration. Therefore, the photosynthetic rate of vegetation changes and the radial increment rate of vegetation become slower or can even be inhibited. Strong transpiration increased trunk shrinkage during daytime under conditions of high temperature and low rainfall, which inhibit the radial growth of trees [48].

On a growth period time resolution, we found the variation of growth limiting factors from temperature in the spring and early summer to precipitation in summer and autumn. This seasonal variation of growth limiting factors of coniferous and deciduous species has been widely observed in temperate and subtropical forests in the summer [47,49]. Climate factors mainly influence moisture availability to trees, thereby affecting their stem radial growth [50]. After early June, large amounts of precipitation significantly increased soil moisture, which indirectly alleviated water deficit in the trunk, thus reducing the negative water potentials that favour radial growth [50–52]. Therefore, the effect of water on stem growth exceeded that of temperature. These mechanisms explain why limiting growth factors switched in summer.

### 4.3. Model Curves for Radial Growth

The annual radial growth of Chinese fir was suitable for the Gompertz model, and the coefficient of determination (CD) of the function reached 0.98 (Figure 4). Considering the uncertainty of the

model, the coefficient of variation (CV) of three parameters ($A$, $\mu$, and $\lambda$) simulated by Gompertz's function were appropriate (CV < 0.11). Camarero et al. [4] noted that the Gompertz model function was not appropriate for adapting to the bimodal growth model in the Mediterranean continental climate. Nevertheless, a unified S-shaped Gompertz growth model for Chinese fir was successfully constructed in our research. In subtropical areas, the radial growth of *Pinus taiwanensis* Hayata was fitted to the Gompertz model, which was a bimodal stem radial growing pattern with two peaks [48]. Michelot et al. [18] also successfully constructed an extended Gompertz function of the double-peaks' growth curves of *Quercus petraea* (Matt.) Liebl. in lowland temperate forests. In subtropical regions, a previous study of *Cinnamomum kanehirae* hayata in subtropical climate successfully fitted the Gompertz model for radial growth [14].

In 2018, the radial cumulative increment was the lowest, because the precipitation in June was less and air temperature was higher. However, the climatic factors during the study period were not extreme, nor were the research cycles too short to reveal any climate trend, and there was no obvious effect on the radial growth of trees [14].

## 5. Conclusions

In this study, the stem radial increments of Chinese fir in subtropical China were monitored with MBDs over four years. We found that the periodic monthly increment curve in a year was a double peak, and the maximum growth rate occurred in April and August. The different social statuses have different radial growth periods, dominant and intermediate trees began in March and stopped in November, whereas suppressed trees of class I began in March and stopped in September. Gompertz models were constructed in our study, and the fitting values of mean monthly increment were basically consistent with the observed values. The sensitivity of radial growth in Chinese fir to temperature was higher than sensitivity to precipitation and relative humidity. The influence of climate factors on tree radial growth shows seasonal variation. In the future, detailed studies on cambial phenology and wood formation of Chinese fir are needed to further elucidate the response mechanism of radial growth to subtropical environmental constraints.

**Author Contributions:** Idea and experimental design, X.D.; experimental implementation, Z.Z., and Y.H.; statistical data analyses, X.D., Y.H., W.X., S.O., and P.L.; manuscript writing, X.D., and Y.H.; discussion and provision of amended advice, W.X. and W.Y. All authors have read and approved the content of the manuscript.

**Funding:** This work was funded by The National Key Research and Development Program of China (2016YFD0600303).

**Acknowledgments:** We would like to thank The National Key Research and Development Program of China (2016YFD0600303) for its financial support. We also thank Yang Yungui, Lu Chengjing and Huang Jingping for sample data measurement, and climate data collection. Finally, we would like to thank Wu Anchi and Zhou Suer for their valuable comments on the revision of this article.

**Conflicts of Interest:** The authors declare no conflict of interest.

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
