# Peer review of "Monthly Radial Growth Model of Chinese Fir (Cunninghamia lanceolata (Lamb.) Hook.), and the Relationships between Radial Increment and Climate Factors"

_forests, doi:10.3390/f10090757_

Round 1

Reviewer 1 Report

GENERAL COMMENTS:

I found this research interesting for better understanding of Chinese fir growing at the plantations in subtropical climate in China. In general, I think that the entire article is well organized and written from the introduction paragraph through the methods, results and discussion, but the authors should pay more attention to the writing itself, using already established phrases and terms in forestry, for example, “tree radius growth” is better to say / write “radial growth of tree”.

SPECIFIC COMMENTS:

line 22: “monthly for four years” correct to “monthly during four years

lines 22-23: “using manual band dendrometers (MBD)” Use only the abbreviation here “MBD”.

lines 47-48: “Alterations in the width of tree rings are known to potentially encode important environmental factors, such as air temperature, precipitation, and other climatic factors.“

Delete “potentially” because the tree rings do encode environmental factors. One more comment, “such as air temperature, precipitation are environmental or climatic factors? I suggest to re-write the whole phrase.

line 70: Correct “tree radius growth” to “radial growth of tree”

line 72: Correct “species of conifer” to “conifer tree species”

lines 72-73: What is the relationship between high quality timber and human society and environment? Please, explain in better.

lines 75-76: “ For growth study in a subtropical region, tree radius growth is an important factor in understanding the climate-growth relationship”. Please, explain, write this phrase better.

line 77: Correct “have mostly focused” to “have been mostly focused”.

line 81-82: Are “temperature, precipitation, and relative humidity” climate indices od climate data?? Please, clarify.

lines 93-94: “annual precipitations ranged from 931 mm to 1724 mm between 1951 and 2018” are this values min and max average precipitation for the period 1951-2018? Explain it better.

line 100: Why site A was reforested and site B remained undisturbed?

line 109: Delete this part “which were installed in trees at breast height (DBH, 1.3 m).” and use only “at DBH”.

line 111: How could you measure “length of each tree ring within 0.01 mm accuracy on the first day of every month.” ? When you work with dendrometer you don’t see tree rings, you can only measure the variation of stem circumference. Please, clarify it.

line 116 Correct “were considerable” to “were considered or were used”

lines 119-120: It would be interesting to show / compare in the same graph long period average climatic conditions 1951-2018 with the conditions during your experiment 2015-2018.

line 127: Correct “were showed” to “are showed”.

line 127-128: “The ages of sample trees were different in the two sites, and site A and site B were 22 and 30 years in 2018, respectively.” Explain this phrase better.

line 307: Correct “dynamism” to “dynamics”

line 321: Move Figure 7 and its corresponding explanation to the results chapter, no to the discussion.

Reviewer 2 Report

The manuscript is rather interesting and provides a new understanding of tree plantations in subtropical Asia.
Some general and several minor improvements are needed:

Although the is generally well written the broader description of the local environment and climate is missing.
Line 68-78: omit the vague, overgeneralized information and provide detailed information about local natural vegetation and ecological differences of fir plantation in comparison with a natural forest of the region.
You can decide to add this information here or in “Study site” (line 90).
Line 96: provide the name of the soil according to the international terminology. Information about pH and predominance of clay in way too inaccurate to allow the reader to identify the soil of studied sites. Is the soil identical on both sites?
Line 97: the methodology of defining the social status of the trees is totally missing. What technique was employed? This is one of the weakest sides of the manuscript.
Additionally, the coding of the groups is very confusing. Modify it in the way that site and status is included, for instance, as1, as2, and bs1 being suppressed trees from site A and B.
Line 101: Rephrase that sentence, the verb is missing. 
Line 122, Fig. 2: include the graph presenting the same climate parameters for loner period (30years) providing the information about the location of the climate station.
Line 216, 226, Fig. 5: The information about what kind of climate data was used is confusing. In the text, authors talked about monthly values whereas in the caption of fig. 5 they spoke of daily data. This problem is present also in other paragraphs and captions. Always be precise with terms of the parameters, don’t use shortcuts like “temperature”, use full-term names e.g., a monthly average of maximum temperature. Correct this mistake in the lines pinpointed here and in whole reminded text.
Line 362: Be more specific, what part of results, and how they can improve the management. Otherwise, remove the sentence from the manuscript.

Author Response

This manuscript is a resubmission of an earlier submission. The following is a list of the peer review reports and author responses from that submission.